# New Variant in Placophilin-2 Gene Causing Arrhythmogenic Myocardiopathy

**DOI:** 10.3390/genes13050782

**Published:** 2022-04-27

**Authors:** Fiama Caimi-Martinez, Guido Antoniutti, Rocio Blanco, Bernardo García de la Villa, Nelson Alvarenga, Nancy Govea-Callizo, Laura Torres-Juan, Damián Heine-Suñer, Jordi Rosell-Andreo, David Crémer Luengos, Jorge Alvarez-Rubio, Tomás Ripoll-Vera

**Affiliations:** 1Inherited Heart Disease Unit, Hospital Universitario Son Llatzer, 07198 Palma de Mallorca, Spain; fiamacaimi91@gmail.com (F.C.-M.); guidoantoniutti@hotmail.com (G.A.); dra.rocioblanco@gmail.com (R.B.); dcremer@hsll.es (D.C.L.); jalvarezr@hsll.es (J.A.-R.); 2Cardiology Department, Hospital de Manacor, 07500 Manacor, Spain; bgvilla@hotmail.com; 3Cardiology Department, Clínica Quirón Palmaplanas, 07010 Palma de Mallorca, Spain; nelson.alvarenga@quironsalud.es; 4Molecular Diagnostics and Clinical Genetics Unit, Hospital Universitario Son Espases, 07120 Palma de Mallorca, Spain; nancy.govea@ssib.es (N.G.-C.); laura.torresjuan@ssib.es (L.T.-J.); damian.heine@ssib.es (D.H.-S.); jordi.rosell@ssib.es (J.R.-A.); 5Health Research Institute of the Balearic Islands (IdISBa), 07120 Palma de Mallorca, Spain; 6CIBER of Physiopathology of Obesity and Nutrition, 28029 Madrid, Spain

**Keywords:** cardiomyopathies, CVD genetics, NGS for diagnostics of CVDs-

## Abstract

Introduction: Arrhythmogenic cardiomyopathy (ACM) is an inherited disease characterized by progressive fibroadipose replacement of cardiomyocytes. Its diagnosis is based on imaging, electrocardiographic, histological and genetic/familial criteria. The development of the disease is based mainly on desmosomal genes. Knowledge of the phenotypic expression of each of these genes will help in both diagnosis and prognosis. The objective of this study is to describe the genotype–phenotype association of an unknown *PKP2* gene variant in a family diagnosed with ACM. Methods: Clinical and genetic study of a big family carrying the p.Tyr168* variant in the *PKP2* gene, in order to demonstrate pathogenicity of this variant, causing ACM. Results: Twenty-two patients (proband and relatives) were evaluated. This variant presented with high arrhythmic load at an early age, but without evidence of structural heart disease after 20 years of follow-up, with low risk in predictive scores. We demonstrate evidence of its pathogenicity. Conclusions: The p.Tyr168* variant in the *PKP2* gene causes ACM with a high arrhythmic load and with an absence of structural heart disease. This fact emphasizes the value of knowing the phenotypic expression of each variant.

## 1. Introduction

ACM is an inherited disease characterized by progressive fibrofatty replacement of cardiomyocytes resulting in ventricular arrhythmias and advanced stages of heart failure [1]. It was first described in 1763 by Giovanni as arrhythmogenic right ventricular dysplasia, although its official publication occurred in 1982; this was later replaced by arrhythmogenic cardiomyopathy [2]. It is currently recorded in 4% to 7% of sudden deaths (SD) [3]. The way of understanding the disease has changed in parallel with the progression of diagnostic studies, which has caused its definition to vary and is currently based on the sum of imaging, electrocardiographic, histological and genetic/familial variables [4,5,6], updated through the Padua criteria, which include organic involvement of the left ventricle [7].

At the same time, classification forms were described according to their clinicopathological presentation based on non-invasive studies relating to the predominance of left/right ventricular involvement and according to the evolution of the disease in stages: (1) the occult phase where there is no structural evidence of disease, but already carrying the risk of SD; (2) the electrical phase characterized by electrocardiographic changes, negative T waves, early repolarisation and ventricular tachycardia (VT); and (3) the structural phase where there is a progressive structural modification in one or both ventricles [8,9].

With respect to its genetic basis, it is associated with genetic variants of autosomal dominant inheritance classified as strong, moderate and limited according to the evidence of association with the disease [10]. Those with evident associations are related to genes that transcribe desmosomal proteins: *PKP2* (reported in up to 46% of the variants presented), *DSP*, *DSC2*, *TMEM43*, *DSG2* and *JUP*; moderate association *DES* and *PLN*; and limited in the case of *LMNA*, *SC5NA*, *CDH2*, *CTNNA3*, *TGFB3*, *TTN* and *MYH7*, among others [1]. On the other hand, syndromic forms associated with recessive inheritance of variants in the *DSP* gene, such as Naxos and Carvajal, should be considered, which present concomitantly with skin lesions and sometimes with characteristic woolly hair. In autosomal dominant forms (the majority), about 50% of affected patients have a family history of the disease and approximately 3% to 6% have more than one pathogenic or probably pathogenic variant contributing to the phenotype [10,11].

Knowledge about the relationship and impact of a genetic variant on the evolution of the disease is crucial, given that there are a few variants with the low presence of structural heart disease and a high risk of malignant arrhythmias, such as variants in *TMEM43*, truncations in *DSP*, lamin A/C and *PLN*, among others. In the presence of these variants, individual risk stratification for each is advocated [12,13].

Patients with ACM are usually oligosymptomatic or asymptomatic. They may present with syncope secondary to episodes of ventricular arrhythmia, palpitations or even sudden death. An atypical symptom is a precordial pain, which could be interpreted in the context of arrhythmia or the so-called hot period of the disease, characterized by precordial pain with elevated cardiac enzymes and normal coronary arteries, in young patients (mean age 14 years) in which the pathophysiological basis could be due to necrosis preceding fibrofatty replacement and possibly to the inflammatory mechanisms involved [13,14,15].

In turn, the annual risk of developing ventricular arrhythmias is 10%, including sudden death, accounting for 13–14% of SCD in athletes [16,17]. Therefore, the recognition of a related pathogenic variant is a fundamental point in the diagnostic and prognostic confirmation that complements, along with cardiac magnetic resonance (CMR), the risk stratification of sudden death calculated by ARVC risk. (Figure 1) [18,19].

Risk stratification considers potential predictors of ventricular arrhythmia: age, sex, cardiac syncope in the previous 6 months, non-sustained VT, number of premature ventricular complexes in 24 h, number of leads with T-wave inversion and right ventricular ejection fraction. The addition of parameters such as assessment of cardiac fibrosis by CMR, the presence of pathogenic genetic variants and the behavior of the rest of the family group will be useful to estimate the natural history of the disease [20,21].

The approach to the disease is based on the prevention of ventricular arrhythmia on the one hand, through risk stratification and limitation of adverse factors, such as competitive sport. The restriction of physical activity is based on the fact that it is a factor that increases the penetrance of the disease, promotes arrhythmias and accelerates ventricular dysfunction [22].

On the other hand, it is necessary to consider the preventive treatment of SD, which includes antiarrhythmic drug therapy, ablation and defibrillator implantation (ICD). Current efforts are focused on obtaining therapies aimed at delaying cardiac remodeling and the progression of the disease. Focusing on its desmosomal genetic basis, it was suggested that it could be a disease with multisystem involvement, given the fact that all cells in the body have desmosomal proteins in their intercellular junction [23]. Although more data are needed, there is already evidence in mice with objective treatments on desmosomes [24].

We described a new variant in the *PKP2* gene (genomic reference NG_009000.1), which is a gene located on chromosome 12 (p11.21) that is composed of 14 exons, encoding the Placophilin-2 protein. It is the most commonly associated gene with arrhythmogenic cardiomyopathy [25,26].

Placophilin-2 is a 98 kDa protein that is part of the desmosomes, a structure specialized in intercellular junctions and the mechanical resistance of the cell, mainly in those tissues exposed to stretching and friction, such as the myocardium and epithelium [27,28].

Other functions of Placophilin were postulated, such as its involvement in fibrosis and adipogenesis, where it plays a role in the activation of fibroblasts and the promotion of fibrofat deposits through the increased expression of transforming growth factor β (TGFβ). TGFβ triggers a collagen synthesis-promoting cascade. It is also involved in the functioning of sodium and calcium channels, so a variant can cause life-threatening arrhythmias, even in the absence of structural involvement [29,30,31].

## 2. Objectives

To describe the genotype–phenotype association of a PKP2 loss-of-function variant to early arrhythmia in the concealed phase of ACM.

## 3. Materials and Methods

Based on the findings of next-generation sequencing (NGS) in the index patient of a *PKP2* variant, a cascade study of the family group was performed. The presence or absence of the *PKP2* variant was evaluated in 22 patients (the proband and their family members), including 4 patients with SD of unknown cause for whom we have no data.

## 4. Results

The index patient (IV.17) (Figure 2) of Caucasian ethnicity, with no known family history of pathology, started the study at the age of 28 years after an episode of loss of consciousness associated with exercise classified as vasovagal syncope, an event that was repeated 2 years later. The study of this last episode showed self-limited paroxysmal atrial fibrillation, with no abnormal findings in the complementary tests (electrocardiogram in sinus rhythm, narrow QRS and negative T waves in V1 and flattened in V2–V3), ergometry and echocardiogram (Figure 3). He was asymptomatic with no intercurrences until he was 34 years old when he presented with precordial pain and palpitations during sports activities. He stopped the sporting activity he was doing and went to the emergency department where sustained VT without hemodynamic decompensation was recorded. After a coronary and echocardiographic study, which was structurally normal, he underwent an electrophysiological study in which episodes of polymorphic sustained VT were induced by isoprenaline infusion, and VT ablation was subsequently performed. At the time of evaluation, his risk stratification was estimated using the risk calculator, resulting in 3.3% at 5 years of ventricular arrhythmia and SD.

CMR was performed without pathological findings (Figure 4). Subsequently, ICD implantation was performed as the primary prevention of SD.

In the context of a patient with sustained ventricular arrhythmia with repeated syncope and the absence of structural disease, a genetic study was performed by massive NGS sequencing, which identified a heterozygous variant in *PKP2*: NP_004563.2:p.Tyr168*/NC_000012.11:g.33031310G>T that had been previously described in only one patient. There have been 927 *PKP2* variants described in public databases, among which the p.Tyr168* variant, found in the presented family group, is not found [32]. This variant is a nonsense variant, located in exon 3 of the gene and affects the N-terminal end of the protein; this means that if the aberrant transcripts were translated, they would produce a truncated protein and a variable loss of structure. We found an unknown frequency in Varsome, Clinvar, Homologene, Mutation testing and genoma. However, there is a publication on the presence of the p.Tyr168* variant in a single affected person, within an Iranian family, who was diagnosed with ACM with evident structural heart disease and biventricular involvement at the time of diagnosis and his carrier mother without familiar screening [33].

Regarding its pathogenicity, we classified it as pathogenic according to the variant classification criteria of the American College of Genetic Medicine and Genomes and Molecular Pathology Association consensus [34]. Its pathogenicity is due to the following criteria:

PVS1: Encodes a stop codon, obtaining a non-functional protein.

PM2: Not found in general population controls.

PP3: According to in silico studies.

With these results, through the sustained ventricular arrhythmia and probably pathogenic variant, the diagnosis of ACM is made.

A cascade study of the family group was initiated. His sister (IV.16), 37 years old, was asymptomatic at the time of evaluation, with complementary studies ECG: sinus rhythm, narrow Qrs, with negative T waves V1–V2 and flattened V3, echocardiogram and CMR without pathological findings (Figure 5 and Figure 6). She is a carrier of the *PKP2* p.Tyr168* heterozygous variant. During follow-up, at 43 years of age, she presented an episode of cardiogenic syncope with subsequent evidence of frequent ventricular extrasystoles and episodes of non-sustained VT, so it was decided to implant an ICD for primary prevention, with estimated risk stratification of 1.9% at 5 years. The patient presented appropriate discharges less than one year after implantation.

At the same time, the son of (IV.16), V.10, was evaluated as a carrier of the same variant.

His mother (III.13), who was asymptomatic at the time of screening, had a positive genetic result for the heterozygous variant in *PKP2* p.Tyr168*, presenting the following ECG studies: sinus base rhythm T negative V1–V2 and flattened V3, echocardiogram and CMR without pathological findings of relevance. His calculated risk stratification was 0.9% (Figure 7 and Figure 8).

The proband’s daughter (V.12) is also a carrier of the *PKP2* variant, and presents ECG, Doppler echocardiogram and 24 h Holter without pathological findings, with a risk stratification estimated at 0.3%.

The family tree revealed four patients with sudden cardiac death and unremarkable past medical history. However, we have no further information about these patients. The other 17 relatives were studied with EKG, echo and genetic testing without pathogenic findings.

## 5. Discussion

The evidence of the pathogenicity of this variant in the family group affected by ACM without evidence of structural involvement and the presence of malignant arrhythmias in middle age in an oligosymptomatic context has allowed us to systematically search among the relatives of the index case for the presence of the variant, with diagnostic and prognostic utility; this emphasizes the clinical surveillance of those phenotypically negative carriers who do carry the genetic variant, which involves the real value of the genetic study [35].

## 6. Conclusions

There is a clear need to describe the genotype/phenotype association of new variants found, so that we can recognize those variants with a high risk of ventricular arrhythmia, even without evidence of structural heart disease, especially in the case of ACM. This contributes to the identification of at-risk carriers. The p.Tyr168* variant in *PKP2* is associated with ACM and presents in the family described with a phenotype that is exclusively malignant ventricular arrhythmias as the first symptom, which precedes any structural myocardial involvement in 20 years of follow-up.

## Figures and Tables

**Figure 1 genes-13-00782-f001:**
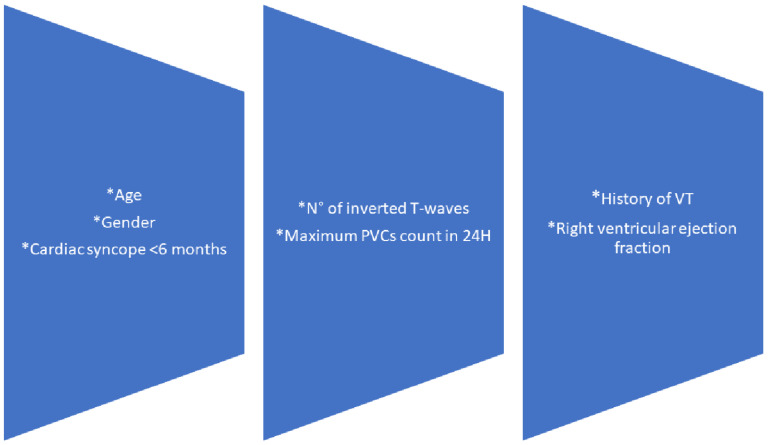
Arrhythmogenic cardiomyopathy sudden death risk calculator. Adapted from ARVC risk. N°: number, PVCs: premature ventricular complex, H: hours, VT: Ventricular tachycardia.

**Figure 2 genes-13-00782-f002:**
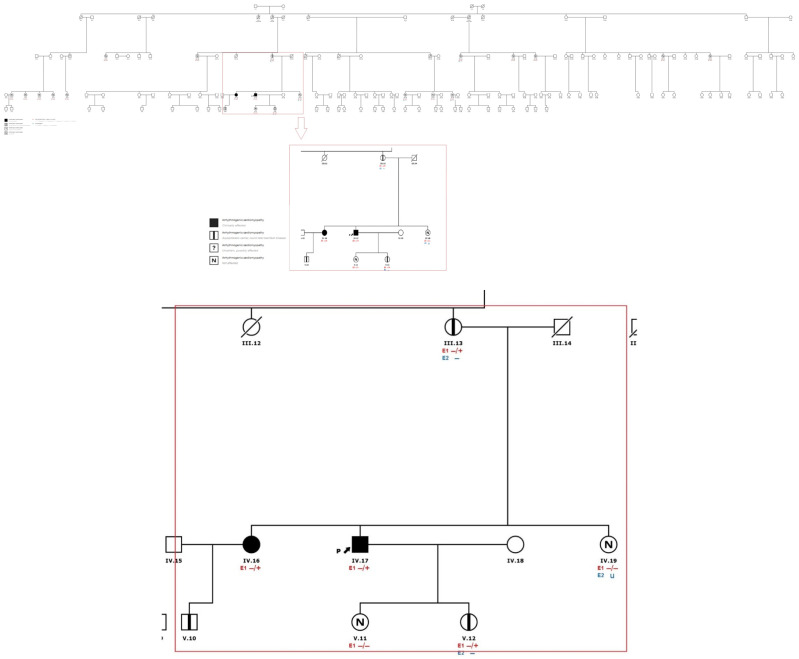
Family tree 
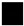
 arrhythmogenic cardiomyopathy—clinically affected. 

 arrhythmogenic cardiomyopathy—asymptomatic carrier, could later manifest disease. 
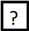
 arrhythmogenic cardiomyopathy—uncertain, possibly affected. 
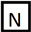
 arrhythmogenic cardiomyopathy—Not affected. E1 Genetic study: (− negative)/(+ positive). E2 Echocardiogram: (− negative)/(+ positive).

**Figure 3 genes-13-00782-f003:**
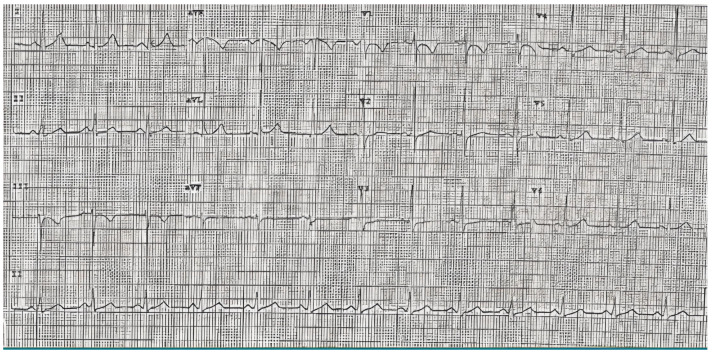
Ambulatory electrocardiogram of index patient IV.17.: Sinus rhythm, narrow QRS and negative T waves in V1 and flattened in V2–V3.

**Figure 4 genes-13-00782-f004:**
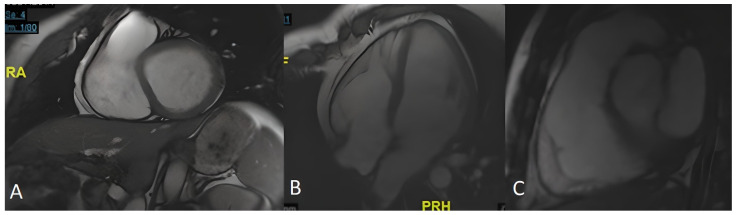
Cardiac CMR of index patient IV.17. The detailed study of the RV shows a non-dilated cavity, with no apparent abnormalities of the parietal contour and no evidence of fatty infiltration. The global dynamics of the right ventricle are preserved, with no alterations in segmental motility. There is no late enhancement in myocardial suppression sequences suggesting necrosis/fibrosis. (**A**) short axis; (**B**) four chambers; (**C**) right ventricle.

**Figure 5 genes-13-00782-f005:**
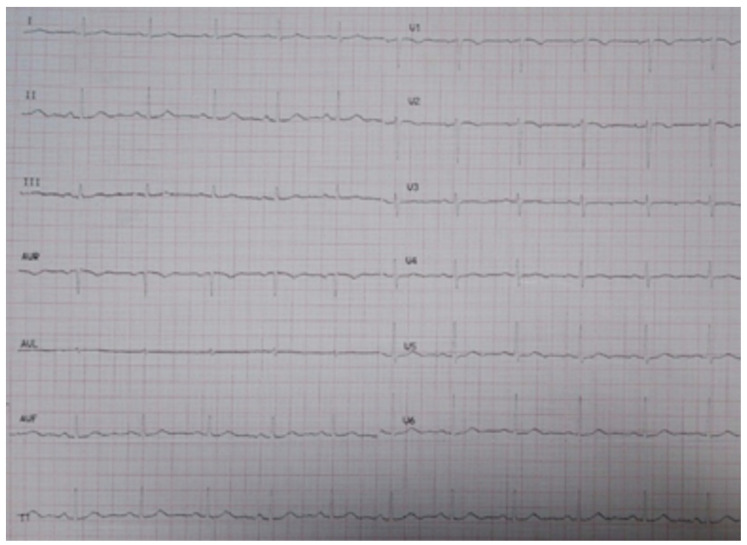
Electrocardiogram patient IV.16. Sinus rhythm, narrow Qrs, with negative T waves V1–V2 and flattened V3.

**Figure 6 genes-13-00782-f006:**
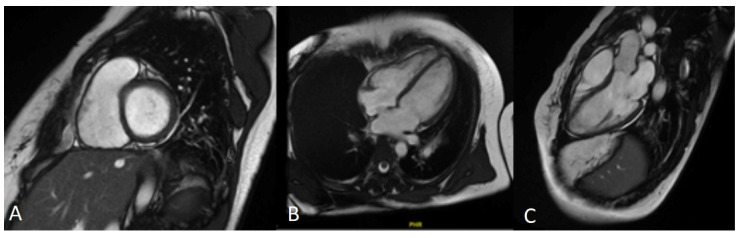
CMR patient IV.16. Right ventricle of normal volumes: end-diastole volume 70 mL/m^2^, end-systole volume 33 mL/m^2^ and normal systolic function, ejection fraction 53%. No dyskinetic areas or focal aneurysmal dilatations were observed. There is no late enhancement in myocardial suppression sequences suggesting necrosis/fibrosis. (**A**) short axis; (**B**) four chambers; (**C**) long axis, four chambers.

**Figure 7 genes-13-00782-f007:**
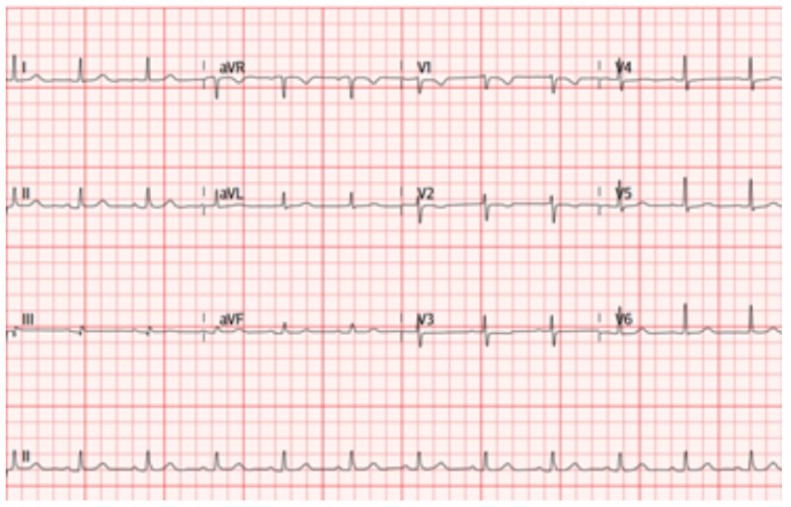
Electrocardiogram patient III.13. Sinus base rhythm T negative V1–V2 and flattened V3.

**Figure 8 genes-13-00782-f008:**
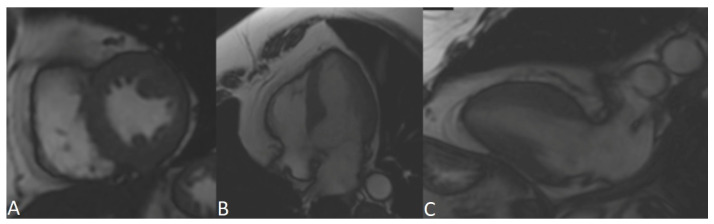
CMR patient III.13 Right ventricle with normal volumes: end-diastole volume 44 mL/m^2^ and end-systole volume 21 mL/m^2^ with preserved global systolic function, ejection fraction 52%. No late gadolinium enhancement. Four chambers; (**A**) Short axis; (**B**) Four chambers (**C**) Long axis.

## Data Availability

Not applicable.

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
