# Peer review of "New Variant in Placophilin-2 Gene Causing Arrhythmogenic Myocardiopathy"

_genes, 2022, doi:10.3390/genes13050782_

Round 1

Reviewer 1 Report

In this study, Caimi-Martinez et al., identified a new heterozygous variant in the desmosomal protein Placophilin-2 gene (PKP2) in a family. The authors described this variant as correlated with the arrhythmogenic cardiomyopathy (ACM).

In my opinion, the manuscript has some concerns:

  • The authors claimed that the mutation “had not been previously described” (line 143), then they said “there is a publication on the presence of the p.Tyr168* variant in a single affected person” (line 149). In my opinion, these two sentences are in contrast.

  • In the study, only 5 subject are described. What about the other 17 family members and “4 patients with MS (?) of unknown cause” of the material and methods section?

  • The figures are not well described in the legend. For example, in the figure 2 (family tree) there is not a clear explanation of what the symbols in the pedigree mean. Moreover, the upper panel of the image is not visible. In the figures 3 and 4, the figures 5 and 6 and the figure 7 and 8 there are not any explanations, and it is very difficult for a nonmedical person to understand what the images should show. In the legend of figures 4, 6 and 8 there are some letters (A, B, etc) but I can not see any letter in the figures, what they refer to?

  • Some abbreviations are not described in the text (see line 113 – MS or line 199 – MCA).

  • The link at the line 72 https://arvcrisk.com goes to an unavailable website.

  • In the line 153-155, the authors wrote: “Regarding its pathogenicity, we have classified it as pathogenic according to the variant classification criteria of the American College of Genetic Medicine and Genomes and Molecular Pathology Association consensus”. The authors should describe how they classified the mutation of the study as “pathogenic”, because it is not clear.

Author Response

Dear reviewer:

                                       I appreciate the time taken to make the corrections that will improve my research entitled "New variant in the Placophilin-2 gene causing arrhythmogenic Cardiomyopathy" written by Fiama Caimi- Martinez, Guido Antoniutti, Rocio Blanco, Bernardo Garcia de la Villa, Nelson Alvarenga, Nancy Govea-Callizo, Laura Torres-Juan, Damian Heine-Suñer, Jordi Rosell.Andreo, David Crémer-Luengos, Jorge Alvarez-Rubio, Tomas Ripoll-Vera.

Attached is the work done in response to the requested corrections. We look forward to a prompt response. Thank you very much. Kind regards. Fiama Caimi.

Reviewer 1:

  1. The authors claimed that the mutation “had not been previously described” (line 143), then they said “there is a publication on the presence of the p.Tyr168* variant in a single affected person” (line 149). In my opinion, these two sentences are in contrast.

*We are aware of and declare the existence of a publication about the variant, but we believe that its extension is insufficient, it does not have extensive follow-up time as the one presented. Neither does it present a study of carriers.

It had only been described in the literature an isolated case report of affected patient and his Carrier mother without familiar screening.

  1. In the study, only 5 subject are described. What about the other 17 family members and “4 patients with MS (?) of unknown cause” of the material and methods section?

*We added data about the 17 patients not mentioned, since they turned out not to be carriers..

The family tree revealed 4 patients with sudden cardiac death and unremarkable past medical history. However, we have no further information about these patients. The other 17 relatives have been study with EKG, echo and genetic testing without pathogenic findings.  

  1. The figures are not well described in the legend. For example, in the figure 2 (family tree) there is not a clear explanation of what the symbols in the pedigree mean. Moreover, the upper panel of the image is not visible. In the figures 3 and 4, the figures 5 and 6 and the figure 7 and 8 there are not any explanations, and it is very difficult for a nonmedical person to understand what the images should show. In the legend of figures 4, 6 and 8 there are some letters (A, B, etc) but I can not see any letter in the figures, what they refer to?

*We extended the explanation of the figures

Figure 2: Family tree    arrhythmogenic cardiomyopathy- Clinically affected.

arrhythmogenic cardiomyopathy-Asymptomatic carrier, could later manifest disease

 arrhythmogenic cardiomyopathy-Uncertain, possibly affected

 arrhythmogenic cardiomyopathy- Not affected

E1 Genetic study: (- negative)/ (+ positive)/ (“U” uninformative)

E2 Echocardiogram: (- negative)/ (+ positive)

*About the image above I have sent a note. We have the image in good quality and different formats but also, it is difficult to view due to the large size. So I don't know what would be the best solution.

  1. Some abbreviations are not described in the text (see line 113 – MS or line 199 – MCA).

*Corrected abbreviation mistakes mentioned above.

  1. The link at the line 72 https://arvcrisk.com goes to an unavailable website.

*We know that at times the link does not work so we have decided to remove it to avoid inconveniences.

  1. In the line 153-155, the authors wrote: “Regarding its pathogenicity, we have classified it as pathogenic according to the variant classification criteria of the American College of Genetic Medicine and Genomes and Molecular Pathology Association consensus”. The authors should describe how they classified the mutation of the study as “pathogenic”, because it is not clear.

*We improved the explanation of the ACGM criteria for variant pathogenicity:

Its pathogenicity is due to the following criteria:

PVS1: Encodes a stop codon, obtaining a non-functional protein.

PM2: Not found in general population controls.

PP3: According to in silico studies.

Reviewer 2 Report

Abstract

  • Could extend the abstract to include background, case description, and discussion/conclusion

Introduction

  • Excellent review of relevant background literature

Objectives

  • Consider including the following
    • To describe a new association between a PKP2 loss-of-function variant to early arrhythmia in the concealed phase of ARVC
    • To emphasize the important of genetic testing informing risk of sudden death

Materials an Methods

  • Capitalized the first letter
  • Please define MS

Results

  • Capitalize QRS and T waves
  • Please include CMR volumetric of RV if available and specifically comment on whether RV regional wall abnormalities are present.
  • Please pronoun under paragraph describing proband’s mother from his to her.

Figure 3. Patient IV.17 instead of IV.7

Figure 4. Patient IV.17 instead of IV.7

Figure 6. Patient IV.16 instead of IV.6

Figure 7 and 8. Patient III.13 instead of III.4

Discussion

  • Should change negative carriers who do carry the genetic variant instead of do not.
  • Can move the discussion about concealed phase in intro to here

Conclusion

  • Please define MCA
  • Could reorganize the content to make a stronger take-home point

Author Response

Dear reviewer:

                                       I appreciate the time taken to make the corrections that will improve my research entitled "New variant in the Placophilin-2 gene causing arrhythmogenic Cardiomyopathy" written by Fiama Caimi- Martinez, Guido Antoniutti, Rocio Blanco, Bernardo Garcia de la Villa, Nelson Alvarenga, Nancy Govea-Callizo, Laura Torres-Juan, Damian Heine-Suñer, Jordi Rosell.Andreo, David Crémer-Luengos, Jorge Alvarez-Rubio, Tomas Ripoll-Vera.

Attached is the work done in response to the requested corrections. We look forward to a prompt response. Thank you very much. Kind regards. Fiama Caimi.

  1. Could extend the abstract to include background, case description, and discussion/conclusión

*We have extended the abstract to be more descriptive.

  1. Consider including the following:

* To describe a new association between a PKP2 loss-of-function variant to early arrhythmia in the concealed phase of ARVC To emphasize the important of genetic testing informing risk of sudden death. we have replaced it

  1. Capitalized the first letter.

*We have corrected it

  1. Capitalize QRS and T waves.

*We have corrected it

  1. Please include CMR volumetric of RV if available and specifically comment on whether RV regional wall abnormalities are present.

*We attach the maximum details we have of studies, somewhat old (the MRI of the index case is from 2008). We do not have the numerical ventricular volume. The detailed study of the RV shows a non-dilated cavity, with no apparent abnormalities of the parietal contour and no evidence of fatty infiltration. The global dynamics of the right ventricle is preserved, with no alterations in segmental motility. There is no late enhancement in myocardial suppression sequences suggesting necrosis/fibrosis

  1. Please pronoun under paragraph describing proband’s mother from his to her.

*His mother

  1. Figure 3. Patient IV.17 instead of IV.7 Figure 4. Patient IV.17 instead of IV.7 Figure 6. Patient IV.16 instead of IV.6Figure 7 and 8. Patient III.13 instead of III.4

*We have corrected it

  1. Discussion: Should change negative carriers who do carry the genetic variant instead of do not. Can move the discussion about concealed phase in intro to here.

* We have changed the suggestions in this section

  1. Please define MCA

*We correct the translation mistake

  1. Could reorganize the content to make a stronger take-home point

*We have tried to emphasize the conclusions
